# How Do Women on Board Reduce a Firm's Risks to Ensure Sustainable Performance during a Crisis?

**Rubeena Tashfeen** [1], **Irfan Saleem** [2], **Muhammad Ashfaq** [3,*], **Umara Noreen** [4] and **Muhammad Shafiq** [5]

1    UCP Business School, University of Central Punjab, Lahore 54590, Pakistan; rubeena.tashfeen@ucp.edu.pk
2    Faculty of Business, Sohar University, Sohar 311, Oman; isaleem@su.edu.om
3    Department of Finance and Management, IU International University of Applied Sciences, 53604 Bad Honnef, Germany
4    College of Business Administration, Prince Sultan University, Riyadh 12435, Saudi Arabia; unoreen@psu.edu.sa
5    Department of Project & Operations Management, Islamia University of Bahawalpur, Bahawalpur 63100, Pakistan; m.shafiq@iub.edu.pk
*    Correspondence: muhammad.ashfaq@iu.org

**Abstract:** The research applies the Upper Echelons Theory and the Lehman Sisters Hypothesis to explain how women board members use investment, financial, and liquidity techniques to reduce risk and increase a firm's value. An original dataset of listed US companies is analyzed to show how women strategists contribute to value creation and mitigate stock volatility and bankruptcy. A simultaneous equations approach captures the interplay between a company's use of debt and financial derivatives. According to this research, organizations that employ derivative instruments benefit more from having women in advisory roles because women encourage proactive risk management and develop effective risk control measures. The research implies that businesses should actively promote gender equality on their boards rather than merely recognizing the need for diversity.

**Keywords:** woman risk strategist; upper echelons theory; Lehman sisters hypothesis

## 1. Introduction

The global financial crisis is a critical test of the competence of the corporate board for the firm's long-term survival [1]. The recent COVID-19 pandemic and subsequent economic slowdown and stock market crash serve as a prime example of this [2]. At the same time, classically, the financial crisis of mid-2007 and early 2009 tested the resilience of the corporate boards [3]. So, having a diverse corporate board could help a firm's survival during times of crisis.

There is a significant theoretical discussion and empirical studies about the ability of women to efficiently handle firm performance and risk as a part of top management teams [4–6]. Studies claim that the global financial crisis could have been averted, as women are more risk-averse (e.g., [7,8]). Furthermore, some research agrees that the inclusion of women on the corporate board adds to the firm's value for the stakeholders [9,10] and enhances earning management for shareholders [11,12]. However, other scholars claim that the linkage between having women on the corporate board and a firm financial performance follows a U-shape and creates upheaval among directors [13]. Similarly, a study claims that having a female director on the corporate board decreases the organization's performance [14]. As a result, a firm should expect a relatively low financial return and an overall loss of value for stockholders [13]. Further muddying the waters, research scholars prove that gender diversity has nothing to do with performance and appears endogenous [1,5]. Therefore, this research study uses the Lehman Sisters' Hypothesis and UET to address theoretical, contextual and empirical research gaps to address the latest call for further research [2,3].

The study contributes multiple ways to corporate governance literature using the Upper Echelons Theory [15], i.e., how the top management team uses experiences, knowledge, and values to predict future strategic decisions by introducing gender diversity at the corporate board level [15]. First, this study extends the argument about the Lehman Sisters Hypothesis by testing this theory on listed US firms [16]. In the past, scholars often overlooked the contribution of female directors in corporate board decisions regarding financial derivatives and women's role in the company's sustainability performance [5]. So, this study expands on the existing research and theory regarding the factors that influence derivative decisions. This study also makes a unique contribution by analyzing how gender diversity affects the implementation of risk strategies in financial, investment, and liquidity decisions to avoid bankruptcy. Second, this study highlights the importance of women's involvement in the overall corporate governance structure when it comes to managing a firm's financial risk [17,18]. Third, this study examines gender diversity risk strategies related to financial, investment, liquidity decisions, and strategic involvement (e.g., [19,20] relates to the application of risk strategies (e.g., [8]). In the past, scholars have left a research question unanswered: how can female representation on a corporate board help to reduce risk in a firm's hedging decisions [21]? Lastly, this research provides insights into the association of women directors with the hedging decisions directly and indirectly through their influence on determinants of derivatives: investment growth opportunities, liquidity decisions and financial distress [22]. Overall, studying the board gender diversity and its influence on policy formulation and strategies could be our crucial contribution to the theory, context and research in the integrated areas of strategic and sustainable decision-making, corporate finance and corporate governance.

This research aims to analyze and compare corporations that utilize derivatives with those that do not to ascertain if having female directors results in greater success for companies. In the subsequent sections, a literature review is presented to seek support from the Lehman Sisters' Hypothesis and the Upper Echelons Theory for our hypotheses. Next, the paper delves into the reasoning behind the hypotheses by examining relevant theories. In the third and fourth sections of the study, sample data, variable measurements, and results are presented. In the last section, the study emphasizes the need for further research in the integrated area of corporate governance. It also provides practical insights for practitioners to consider.

## 2. Literature Review

### 2.1. Upper Echelons Theory and Woman Strategist at Corporate Board

Theories such as organizational identity (e.g., [23], resource dependence [23], and agency [24] are mainly invoked to link the gender diversity–performance nexus [19]. But the Upper Echelons Theory (UET) serves as the critical theoretical framework underlying the present study because of its clarity in representing female-performance linkage at the Top Management Team (TMT) level [15]. According to the upper echelons theory, introducing diversity into the top management team in terms of knowledge, skills, and aptitude (KSA) can enhance the strategic decision-making effectiveness of the corporate board, thus contributing to the sustainable performance of the firm [15]. So, it can be claimed that corporate board members differ in their cognitive frames to influence a firm's strategic decision-making [25] due to different sets of KSA. So, the heterogeneous TMT, including female directors, should use KSAs to take effective strategic decisions on behalf of the firm's shareholders to avoid financial loss and minimize the risk of bankruptcy [26]. For instance, governance scholars have used the cognitive frames of board members [19]. Cognitive-frames of any board member refer to their information-seeking practices and risk evaluation processes while making strategic and sustainable decisions [15]. When considering strategic choices to enhance the sustainable performance of a listed company, the TMT must rely on their knowledge, experiences, and values to interpret information. One important factor to consider is diversity, as it can play a significant role in adding value. For this study, researchers believe that male and female board members significantly differ

in their cognitive frames, i.e., information-seeking/evaluation behaviors. Accordingly, they differ in making strategic choices for the firm [19]. Moreover, supporters of the Lehman Sisters Hypothesis also say that gender bias makes it harder for women to reach the top of a financial company, even though those who do tend to perform better under pressure [7]. It appears that women have limited influence on company boards when it comes to strategic decision-making and implementing financial risk mitigation strategies. This lack of representation could potentially contribute to financial crises, such as those resulting from the post-pandemic and Ukraine situations[27]. Thus, the study assumed that the board's heterogeneity is likely to affect a firm's strategic and sustainable decision-making choices in terms of the firm's value, investment strategy and firm's risks under the Lehman Sisters Hypothesis and UET theoretical framework (e.g., [6,7,28]).

*2.2. Hypothesis Development*

Studies have shown that gender stereotypes prevalent in corporate boardrooms pose obstacles for women seeking to ascend to top positions in the financial decision-making of firms [7,24]. Still, many companies are adding women on board, and researchers have found that gender diversity on boards reduces risk, which indirectly ensures a firm's sustainable performance on the stock market (e.g., [21,22], which is consistent with the theorization of UET focusing on the heterogeneity of boards concerning gender [28]. The reason for adding Women on board is that they are more risk-averse than men [17]. So, it can be assumed that there is a positive association between female directors on corporate boards and firm bankruptcy risk [5]. Therefore, it is hypothesized:

**Hypothesis 1a (H1a).** *Gender heterogeneity on the corporate boards reduces risk in US-listed firms that use derivatives to ensure sustainable performance.*

The primary role of the corporate board is to safeguard and advance the strategic interests of the shareholders by managing financial investments, ensuring the company's longevity and expansion, and providing disciplinary oversight [19]. The Upper Echelons Theory (UET) suggests that executives' experiences, beliefs, and personalities play a crucial role in shaping their perception of situations and ultimately influencing their decision-making process [15]. Therefore, it is important to have diversity on a corporate board to monitor managers under UET assumptions. Without it, there is a risk of financial loss for the company [2,15]. Further, women are naturally risk-averse and therefore do not take unnecessary financial risks that could ensure the sustainable performance of the firm [6]. A study [4] shows that men, in comparison to women, trade 45% more due to overconfidence. Additionally, research indicates that having a higher proportion of women on corporate boards can lead to decreased financial risk for the company through proficient management of portfolio investments and effective implementation of liquidity and financial strategies [9,29,30]. The preceding argument is supported by numerous studies, including those in psychology and economics (e.g., [2,17,24]. Therefore, the following hypothesis is proposed:

**Hypothesis 1b (H1b).** *Gender heterogeneity on the corporate boards reduces risk through investment, financial and liquidity-related strategies for US-listed firms to ensure sustainable performance.*

Generally, scholars are agreed that board diversity positively impacts a firm's financial performance [10]. According to experts, a positive correlation exists between a company's hedging practices and its inclusive governance structure, ultimately leading to improved governance and increased value. This connection also impacts the company's risk management strategies, including its monetary, investment, liquidity, and reimbursement practices [22]. It can be inferred that a company's value can be enhanced by implementing effective and appropriate financial policies by its board of directors. This highlights the expertise, competence, and skills possessed by the individuals serving on the board [15]. Some researchers argue that having a diverse gender representation on board can positively impact a company's performance, specifically in utilizing financial derivatives effectively [31]. The inclusion of

women in TMT has a positive impact on board strategic decisions. For instance, they tend to use financial derivatives such as futures and options contracts, and credit default swaps due to their inherent risk aversion behavior [12]. Research suggests that having an appropriate number of women directors can improve a firm's reputation with stakeholders by mitigating financial risks [11]. So, we can assume that gender diversity on the corporate board would be essential to hedging decisions when a dominating group determines these on the board. Based on the above discussion and literature support, we propose the following hypothesis:

**Hypothesis 2a (H2a).** *Gender heterogeneity on the corporate boards enhances firm value for listed US firms that effectively use derivatives to ensure sustainable performance.*

The literature supports the idea that gender-diverse corporate boards enhance value. For instance, a study found that women provide a wider range of skills, better customer awareness, and higher levels of corporate governance, favourably affecting a firm's success [19]. Other research suggests that implementing the policy of board gender diversity [23] can strengthen board independence [31], as well as significantly lower the danger of homogeneous decisions and thinking [22]. Corporate board members may utilize financial tactics to evaluate a company's capital structure. These strategies may include analyzing the types and levels of external debt usage, financing expenses, and investment in different derivatives, all at a strategic level to benefit the company they serve [8]. Therefore, effective boards would design financial policies to ensure sustainable performance [9,32,33], increase debt capacity, and lower financial costs through the best use of derivatives [34]. Gender-diverse boards, according to researchers, would offer greater value to the firm by lowering underinvestment costs and enhancing investment growth chances and liquidity through managers' efficient use of derivatives. Therefore, researchers posit that diversity on the corporate board helps monitor through promoting derivatives usage and effective hedging strategies to increase value [22] and reduce risk in firms that employ derivatives to ensure sustainable performance [9,35]. Thus, in respect of firm value through gender diversity in both derivative user and non-user firms, scholars hypothesize the following:

**Hypothesis 2b (H2b).** *Gender heterogeneity on the corporate boards enhances firm value through investment, financial and liquidity-related strategies for US-listed firms to ensure sustainable performance.*

## 3. Data, Sample, Variables and Methodology

### 3.1. Data and Sample

Panel A of Table 1 outlines the sample selection procedure. We gathered data from 2741 non-financial companies listed on the NYSE over the nine years from 2013 to 2021, totalling 24,669 observations. We excluded insurance companies due to their distinct financial characteristics and regulation. After removing firms with missing data, our final sample size is 9605 observations. In Table 2, Panel B presents the industry breakdown for the sample.

**Table 1.** Derivation of sample categorization.

| Panel A: Sample Derivation | |
| --- | --- |
| | **Number of Observations** |
| Sample from NYSE | 24,669 |
| Adjust years with missing data | (2741) |
| Adjust insurance companies | (352) |
| Total sample size before deleting missing values | 21,576 |
| Adjust Missing observations | (2073) |
| The total size of the sample before adjusting governance variables | 19,503 |
| Less missing governance observations | (9898) |
| Final sample | 9605 |

**Table 2.** Industry categories of sample firms.

| Panel B: Industrial Classification of Included US Firms | | |
|---|---|---|
| Industry | No. of Firms | Pctg. of Sample |
| Forestry and Agriculture | 32 | 0.003 |
| Other Business and Professional Services | 1040 | 0.053 |
| Hospitality and Leisure | 301 | 0.048 |
| Health and Education | 263 | 0.019 |
| Real Estate | 99 | 0.480 |
| Information Providers | 847 | 0.132 |
| Service Provider | 1269 | 0.088 |
| Utilities | 463 | 0.010 |
| Construction | 179 | 0.103 |
| Mining, Oil and Gas | 505 | 0.027 |
| Manufacturing | 4607 | 0.031 |
| Total | 9605 | 1.000 |

### 3.2. Measuring Variables

#### 3.2.1. Derivatives Measure

Direct Edgar extraction software is used to retrieve derivatives data from SEC 10-K statements from firms employing derivatives. A highly cited study was used as a reference to employ a methodology for measuring and researching data on derivatives [36]. The scholars used search words such as swap*, derivative*, forward, future and option to filter out derivative users. This study has used an asterisk mark to search for any keywords with these prefixes. After that, the sample of derivative-user firms is obtained by matching the sample CIK numbers to the larger sample through the Direct Edgar filter. This search yields many rows. The final sample of derivatives firms is obtained manually from this data set by reading the sentences.

#### 3.2.2. Risk Measure

In this study, firm risk is measured by stock return volatility. Utilizing market efficiency, stock price volatility is considered a suitable proxy for measuring this risk [36]. In line with academic research, this study utilized the fluctuation in stock returns as an indicator to assess the potential risk associated with the firm's assets and liabilities. The second risk measure is the bankruptcy risk of a listed firm. Accordingly, this research used Z-Score to measure bankruptcy risk [6,37] (See Table 3).

**Table 3.** Variables, measures formulas and definitions.

| Variable | Measure | Definitions [Database/Source] |
|---|---|---|
| DER | Binary | This binary value is set to '1' if a company uses derivatives and '0' if it does not. [Data Source: Direct Edgar] |
| VALUE | Ln | Q = (DEBT + PS + MVE)/TA; TA (Tangible Assets) represents the book value of the firm's total assets (BS-TOT-ASSET), MVE is the product of a total number of common stock shares outstanding (BS_SH_OUT) and the share price of the company (PX_LAST) PS is preferred equity (BS-PFD-EQY), and DEBT represents short-term and long-term debt. [Data Source: Bloomberg] |
| STOCK RETURN VOLATILITY | Ln | This variable represents the natural logarithm of Stock Return Volatility, which is the standard deviation, from one day to the next, of logarithmic price changes. The 260-day price volatility of the day before the current day is equal to the annualized standard deviation of the relative price change of the last 260 trade days' closing prices, given as a percentage for the day before the current day. [Complied from Bloomberg (using a formula)] |
| BANKRUPTCY RISK | Ln | ZSCORE: This shows how likely it is that a company will go bankrupt in the next two years. If the score is more than 3, bankruptcy is rare in a company. Bankruptcy is possible below the value of 1.8. Where: the Z Score is obtained through ALTMAN-Z-SCORE, which is $= 1.4 * (\text{Retained Earnings}/TA) + 1.2 * (\text{Working Capital}/TA) + (\text{Sales}/TA) + 0.6 * (\text{Market Value of Equity}/\text{Total Liabilities})$ [ Data Source: Bloomberg] |
| BDDIVERS | Total directors on board | Female directors: Total number of women on a board at the end of the year. [Data Source: Corporate Library–Company] |
| BDDIVERS-LEV | Number of directors | The variable represents Women Director's influence on debt/financial policies. BDDIVERS-LEV have the same definitions as shown. Where LEV is Leverage. [ Data Source: Corporate Library–Company] |
| BDDIVERS- LIQUIDITY | Number of directors | This value represents Women Director's influence on short-term assets and liquidity policies. BDDIVERS and LIQUIDITY have the same definitions as shown. [Data Source: Corporate Library–Company] |
| BDDIVERS- RnD | Number of directors | This value represents Women Director's influence on investment policies. BDDIVERS and RnD have the same definitions as shown. [Data Source:Corporate Library–Company] |
| CAPEX | Ln | Total assets are computed by total capital expenditure by a company (CAPEX). The Bloomberg database was used to obtain separate numbers for capital spending and total assets. Capital spending is how much a company spends on buying real fixed assets, excluding purchases of investments. Capital expenditures for utility companies include additions to utility plants, funds for shutting down plants, and non-utility capital expenditures (CAPITAL-EXPEND). Total Assets are the sum of all assets, both short-term and long-term, as shown on the Balance Sheet (BS-TOT-ASSET) [Data Source: Bloomberg] |

**Table 3.** *Cont.*

| Variable | Measure | Definitions [Database/Source] |
|---|---|---|
| LEVERAGE | Ratio | This number shows the total debt based on the value of the company. Short-term debt (BS-ST-BORROW) plus long-term debt (BS-LT-BORROW) equals total debt. The market value of the stock (CUR-MKT-CAP), total debt (as shown above), and preferred equity (BS-PFD-EQY) are added together to find the firm's value. [Data Source: Bloomberg] |
| SIZE | Ln | Size is the natural logarithm of Sales/Revenue/Turnover (SALES_REV_TURN), which is the sum of operating revenues minus any changes to gross sales. Sales taxes, discounts, insurance fees, allowances, excise taxes, and value-added taxes (VAT) are all adjustments. This study includes revenues from financial subsidiaries in industrial companies if the consolidation incorporates those subsidiaries throughout the report. This study does not consider inter-company revenue. Revenues from discontinued operations are excluded. This research includes federal or local government subsidies in certain industries (such as transportation or utilities). [Data Source: Bloomberg] |
| BDSIZE | Number of directors | Directors Total. The total number of directors on a particular board (excluding Emeritus or Advisory member roles). [Data Sources: Corporate Library–Company and Bloomberg] |
| LIQUIDITY | Ln | This variable indicates the quick ratio's natural logarithm. Calculated as the ratio of liquid assets to current liabilities (QUICK-RATIO). Where: Liquid Assets = Marketable Securities and Short Term Investments + Accounts Receivable + Cash and Near Cash Items and Notes Receivable. [Data Source: Bloomberg] |
| RnD | Binary | This dummy variable is set to '1' if a corporation spends money on research and development; otherwise, it is set to zero. Where RnD is the entire amount spent on research and development, including RnD in the profit and loss statement and RnD capitalized during the period. This field may only return profit and loss statements if R&D expenditure is not disclosed. Depending on the company's disclosure, this figure may or may not be the net of government grants, subsidies, or tax credits. (IS-RnD-EXPEND). [Data Source: Bloomberg] |
| TLCF | Binary | Tax Loss Carry Forward (TLCF) is represented by this figure. If a firm has a tax loss carried forward, the figure is used as a dummy variable of 1. Otherwise, zero. [Data Source: Compustat] |
| VOL | Ln | This figure depicts the natural logarithm of Stock Return Volatility (VOL), and it is the same as the one stated above. [Data Source: Bloomberg] |
| MSC | Ln | MSC is a dummy variable that represents Multiple Share Classes. If a corporation has two or more common equity shareholder classes, the variable is set to one; otherwise, it is set to zero. [Data Source: Compustat] |
| PB | Ln | The market-to-book ratio is calculated by multiplying the market price per share by the book value per share (PX-TO-BOOK-RATIO).[Dat Source: Bloomberg] |
| TOTAL ASSETS | Ln | Total assets are represented by this figure, which is the book value of total assets (BS-TOT-ASSET). [Data Source: Bloomberg] |
| Sales | Ln | Same as defined under SIZE. [Source: Bloomberg] |
| MV | Ln | MV denotes the market value of equity as measured by market capitalization (CUR-MKT-CAP). [Data Source: Bloomberg] |

### 3.2.3. Value Measure

Following previous studies [29,36,38], this study uses Tobin's Q to figure out how good a firm's value is regarding growth. The model is based on the idea that a company's decision to invest rests on how profitable it is now and how profitable it will be in the near future. According to studies, Tobin's Q is a firm's market value ratio to its assets' replacement cost. A famous study published in finance management advises that the original formula be modified to make the value measure less complicated and easier to calculate. The components of their measure are easily accessible through a company's financial accounts [39]. They are captured as Tobin-Q = (DEBT + PS + MVE )/TA (See Table 3 for the exact computation of variables and sources from where data were extracted).

### 3.2.4. Diversity Measure

The researchers used several proxies for gender diversity, such as a dummy variable if a business has female directors in a given year, or the fraction of female directors [6,29,38] with the corporate board. Accordingly, this study uses proxies, i.e., the total number of women on the board; the percentage of women; and dummy variables for one, two women and three women on the board of directors. This study compared board diversity (BDDIVERS) through gender diversity and used the total number of women directors on the boards each year (See Table 3 for the exact computation of variables and sources from where data were extracted).

### 3.2.5. Measurement of Control Variables

It is important to include control variables to isolate the effects of independent variables on the dependent variable. Failure to do so may result in inaccurate coefficients. Consequently, this study incorporates control variables into each model utilized in the literature [36,38].

### 3.2.6. Control Variables for Value and Risk Models

In the models with value and risk as dependent variables, this study uses the following control variables to follow the reputed studies: For instance, leverage [24]; firm size [34]; z-score [36]; return on asset—ROA [36,38]; research and development: RnD and capital expenditure: CAPEX [38] and liquidity [40].

The outcomes of empirical studies imply that a company's success may be affected by several company factors, such as those related to the company's business and operating environment. Therefore, this study uses numerous control variables. The bigger the debt, the higher the cost of external funding and interest payments, making the financial situation more precarious. This research study also used total debt scaled by size to proxy for leverage,

similar to previous research [24]. According to research, leverage has a negative link with the firm's value and a positive relationship with bankruptcy risk. A company with numerous investment growth prospects can undertake profitable net present value (NPV) initiatives. Following the recommendation by a reputed study [38], this study made use of proxies for investment growth potentials, such as RnD and CAPEX (see Table 3). It anticipates a healthy relationship with the value of the company. Companies that successfully generate bigger profits also offer their shareholders a greater return on investment and vice versa. The increase or loss in income would impact firm value and risk. According to studies, a lower return on assets affects the current year's risk (e.g., [24]. As a result, the research study controlled for profitability using a measure of ROA, as utilized in prior studies (e.g., [36]. The impact of liquidity on limited enterprises has long been debated. In a reputable study, for example, organizations with more cash on hand are less likely to be in financial problems [40]. This is because there is a negative link between cash on hand and financial trouble. On the other hand, some studies show that when a company's finances worsen, it tends to save money from cash inflows and grow its cash reserves. This would be the natural result of not having access to outside funding. Also, companies with a lot of volatility cut their debt, cut their investments, and keep more cash on hand. They also cut their non-cash current assets and cut back on dividends [24]. So, this research study assumes the relationship between liquidity and risk could be negative or positive, depending on how volatile and financially troubled the business is. The quick ratio is a measure of liquidity that shows how cash reserves and company risk are related. There are two ideas about how the size of a business affects it. Some people say that a company's size is a good indicator of how much it will cost to go bankrupt. If this is true, bigger companies would have higher fixed costs and payments, and their bankruptcy costs would be higher. Others think that a company's size indicates its economies of scale. So that smaller companies would gain from larger companies that have economies of scale and more resources. For instance, a study [34] says that bigger companies would have the money to set up a hedging scheme. So, in the research, company size is used as a stand-in for financial stress and resources. Therefore, the results are mixed [29,38,41]. So, this study uses sales as a proxy for firm size. The period of the study largely covers the financial crisis. Our descriptive statistics support this, where derivative users have lower Z-Scores, indicating higher bankruptcy risk. According to a reputable study, the mean Z-Scores for their sample of derivative users are lower than those for non-users [36]. Therefore this research study also controls for bankruptcy risk, besides year-fixed effects.

### 3.2.7. Control Variables for the Derivatives Models

In the models with derivatives as a dependent variable, this study uses the following control variables: TLCF (e.g., [36]); LIQUIDITY (e.g., [34]; RnD (e.g., [24]. For example, debt (LEVERAGE) is the dependent variable in the first equation of the derivatives simultaneous model. The current study also uses control variables for the second equation: VOLATILITY (e.g., [13]); ROA (e.g., [42]). A company with a convex tax structure usually uses derivatives to lower its tax bill [38,42]. When companies report tax loss carry-forwards, it shows that their tax system is not flat. Like other experts, scholars in this study also take tax convexity into account (e.g., [36]). A famous study in finance also shows that firms with more debt are more likely to be in trouble financially [43]. These companies will perform more trading to increase their ability to take on debt and reduce their reliance on high-interest loans from outside sources. Similarly, scholars in this study use the same measures to stand in for firm leverage and predict that higher debt will raise financial distress, leading to a positive relationship between firm hedging activities and leverage. The liquidity number shows how short-term money problems and problems caused by not investing enough affect the business. If a company ran out of cash reserves, it couldn't take advantage of profitable investment possibilities. If a company does not have enough money on its own, the board may decide to use hedging to improve its ability to raise money. In the same way as a previous study, this study uses the quick ratio to stand in for short-term constraints (e.g., [34]) and predict an inverse association between hedging and liquidity. Investment growth possibilities also encourage companies and their directors to use

derivatives so company can invest in profitable projects that help them meet their sustainability goals. Research and development spending is used in studies to determine if a company has investment growth possibilities [38,42] and obtain a positive relationship. So, scholars in this study use a dummy proxy variable to track the company's research and development (RnD) efforts. The loan (LEVERAGE) is the dependent variable in the first equation of the derivatives simultaneous model. For this model, this study uses control variables similar to those used in older works (e.g., [42]). Researchers in this study use stock return volatility (e.g., [13]) and profitability using ROA (e.g., [42]). This study postulated that DER and volatility will be positively linked to a firm's financial leverage, while ROA and RnD will be negatively linked to financial leverage. The firm size indicator was previously described. A 'dummy' variable with the value 1 has been introduced to all models to account for the financial crisis. Similarly, when it comes to industry effects, the energy and utility industry is highly regulated and has, on average, greater firm success than other industries. To accommodate for this, regression equations utilize a "dummy" variable with a value of 1 if the business is energy and utilities and 0 otherwise.

### 3.3. Measurement of Financial, Investment and Liquidity Strategies of Women Directors

Gender heterogeneity on the corporate boards reduces risk through investment-, financial- and liquidity-related strategies for US-listed firms. To test this hypothesis, this study introduces joint variables in the light of older studies [11,17]. To capture board diversity effects through their investment, financial, and liquidity strategies, the scholars use the joint effect of board diversity with investment, with financial leverage and firm liquidity [9,11,17,44]. The functional form of our empirical tests and interpretation of the interaction variables in the models are shown below:

$$Y = Y_0 + y_1 X_1 + y_2 X_2 + y_3 X_1 X_2 + \epsilon$$

$$Y = y_0 + (y_1 + y_3 X_2) X_1 + y_2 X_2 + \epsilon \ldots \tag{1}$$

$$Y = y_0 + (y_2 + y_3 X_1) X_2 + y_1 X_1 + \epsilon \ldots \tag{2}$$

For instance,

$$Board - Diversity(BOD) = X_1$$

$$Leverage(LEV) = X_2$$

$$Value = Y$$

$$Value = Y_0 + y_1 BOD + y_2 LEV + y_3 BODxLEV + \epsilon$$

$$Value = y_0 + (y_1 + y_3 LEV_2) BOD_1 + y_2 LEV_2 + \epsilon$$

The scholars in this study use the above-demonstrated variations of the equations: For instance, BOD impacts value through the increase/decrease in leverage.

## 4. Methodology

Initially, this study examines the impacts of gender diversity on a firm's derivatives usage decisions. We employ the Simultaneous Equations Model for this purpose. A study describes how borrowing influences the firm's hedging decisions [13]. According to the author, the company's investors occasionally barter short-term debt for long-term obligations. Additionally, they may exchange a long-term loan for a short-term obligation. Therefore, businesses would swap to obtain higher interest rates and reduce borrowing costs. So, based on a study, the simultaneous equations methodology is shown below (see Equations (1) and (2)) to demonstrate how the firm's capital structure and hedging choices happen at the same time [45]:

$$LEVERAGE = b_0 + b_1 DER * + b_2 X_1 + y_1 \ldots \tag{3}$$

$$DEV = c_0 + c_1 LEVERAGE* + c_2 X_2 + y_2 \ldots \tag{4}$$

where LEVERAGE* is the continuous endogenous variable in Equation (1) and DER* is the dichotomous endogenous variable in Equation (2), where one is utilized if DER > zero and zero otherwise.

The independent exogenous variables are shown as $X_1$ and $X_2$ in Equations (1) and (2); $b_2$ and $c_2$ are the parameters in (1) and (2), respectively; $b_1$ and $c_1$ are the parameters of the endogenous variables and $y_1$ and $y_2$ are the error terms of Equations (1) and (2), respectively. Equation (1) illustrates the simultaneous effect of capital structure on derivatives decisions. $X_1$ includes the following control variables: growth, size, volatility, derivatives, and profits. Equation (2) uses a probit-regression model as the binary dependent variable (i.e., DER), and $X_2$ represents the explanatory variables for corporate governance and controls. DER* and LEVERAGE* are predicted values derived from the other equation. Scholars in this study use a well-known probit model by a statistician [46] to extract the simultaneous equations test results. The model differs from others in that it can handle simultaneous equations, with one endogenous variable being binary and the other being continuous. Other models need both endogenous variables to be continuous. However, this model cannot handle joint effects of the predicted variable from the first regression; for example, the predicted value of leverage obtained from the first equation with board diversity (BDDIVERS-LEV). Therefore, the two-stage least squares method is employed with DER and LEVERAGE as the endogenous variables, as suggested by some researchers, to overcome this problem [6,47]. Secondly, this study examines the impacts of board diversity on firm value and risk. The scholars in this study use the ordinary least square (OLS) regression model for this purpose. Lastly, researchers in this paper use the OLS regression methodology to investigate the impacts of board diversity strategies on firm value and firm risk through joint effects of board diversity with investment strategies (BDDIVERS-RD), financial strategies (BDDIVERS-LEV), and liquidity strategies (BDDIVERS-LIQUIDITY).

## 5. Tests and Results

### 5.1. Descriptive Statistics

Table 4 presents the descriptive summary of the dependent variables: derivatives, value, stock return volatility and bankruptcy risk alongside a summary of corporate governance variables. Statistics for board size (BDSIZE) show the lowest (highest) and a mean of 8.82 board members, while a percentile of six members (13). Most boards have one female member, demonstrating that board diversity (BDDIVERS) does not matter. Some boards lack women. Boards include two women at the 75 percentile and three at the highest percentile.

**Table 4.** Descriptive statistics: Panel A: full sample.

| Panel A: Full Sample | N | Mean | Median | SD | Percentiles | | | |
|---|---|---|---|---|---|---|---|---|
| | | | | | 5 | 25 | 75 | 95 |
| Dependent Variables | | | | | | | | |
| VALUE (Ln) | | | | | 9605 | 0.29 | 0.20 | 0.61 |
| VOLATILITY (Ln) | 9605 | 3.76 | 3.75 | 0.48 | −0.55 | −0.12 | 0.65 | 1.41 |
| BANKRUPTCY RISK (Ln) | 9605 | 1.32 | 1.37 | 0.91 | 2.98 | 3.44 | 4.07 | 4.56 |
| Corporate Governance Variables | | | | | −0.05 | 0.82 | 1.81 | 2.72 |
| BDSIZE (no.) | | | | | 9605 | 8.82 | 9.00 | 2.17 |
| BDDivers (no.) | 9605 | 0.97 | 1.00 | 0.97 | 6.00 | 7.00 | 10.00 | 13.00 |
| Control Variables | | | | | 0.00 | 0.00 | 2.00 | 3.00 |
| ROA (as) | | | | | 9605 | 1.38 | 2.30 | 2.32 |
| LIQUIDITY (Ln) | 9605 | 0.21 | 0.18 | 0.88 | −3.82 | 0.95 | 2.90 | 3.57 |
| LEVERAGE (ratio) | 9605 | 0.20 | 0.14 | 0.20 | −1.22 | −0.30 | 0.72 | 1.72 |
| CAPEX (ratio) | 9605 | 0.05 | 0.03 | 0.06 | 0.00 | 0.02 | 0.31 | 0.61 |
| RnD (dummy) | 9605 | 0.50 | 0.00 | 0.50 | 0.01 | 0.02 | 0.06 | 0.16 |
| TLCF (dummy) | 9605 | 0.61 | 1.00 | 0.49 | 0.00 | 0.00 | 1.00 | 1.00 |
| MSC (dummy) | 9605 | 0.09 | 0.00 | 0.29 | 0.00 | 0.00 | 1.00 | 1.00 |
| SIZE (Ln) | 9605 | 6.98 | 6.99 | 1.90 | 0.00 | 0.00 | 0.00 | 1.00 |

The descriptive statistics for the derivative user and non-user firms are displayed under Table 5 Panel B, and for all variables, the difference in means between the two groups is substantial.

**Table 5.** Derivative users and non-users.

| | DER USER | | | NON USER | | | Difference in Means | Wilcoxon *p*-value |
|---|---|---|---|---|---|---|---|---|
| Variables | N | Mean | Median | N | Mean | Median | | |
| Dependent Variables | | | | | | | | |
| VALUE | 4609 | 0.21 | 0.12 | 4996 | 0.37 | 0.28 | −0.16 | 0.000 |
| VOLATILITY | 4609 | 3.71 | 3.69 | 4996 | 3.80 | 3.79 | −0.08 | 0.000 |
| BANKRUPTCY RISK | 4609 | 1.23 | 1.29 | 4996 | 1.40 | 1.46 | −0.17 | 0.000 |
| Corporate Governance Variables | | | | | | | | |
| BDSIZE | 4609 | 9.18 | 9.00 | 4996 | 8.49 | 8.00 | 0.69 | 0.000 |
| BDDIVERS | 4609 | 1.07 | 1.00 | 4996 | 0.88 | 1.00 | 0.19 | 0.000 |
| Control Variables | | | | | | | | |
| ROA | 4609 | 1.52 | 2.29 | 4996 | 1.26 | 2.32 | 0.26 | 0.706 |
| LIQUIDITY | 4609 | 0.05 | 0.05 | 4996 | 0.36 | 0.32 | −0.30 | 0.000 |
| TOTAL ASSETS | 4609 | 2.10 | 2.11 | 4996 | 1.99 | 1.99 | 0.11 | 0.000 |
| LEVERAGE | 4609 | 0.24 | 0.19 | 4996 | 0.16 | 0.09 | 0.08 | 0.000 |
| CAPEX | 4609 | 0.06 | 0.04 | 4996 | 0.05 | 0.03 | 0.01 | 0.000 |
| RnD | 4609 | 0.49 | 0.00 | 4996 | 0.50 | 1.00 | −0.01 | 0.308 |
| TLCF | 4609 | 0.59 | 1.00 | 4996 | 0.63 | 1.00 | −0.05 | 0.000 |
| MSC | 4609 | 0.10 | 0.00 | 4996 | 0.09 | 0.00 | 0.00 | 0.689 |
| MV | 4609 | 7.48 | 7.41 | 4996 | 6.97 | 6.81 | 0.51 | 0.000 |
| SIZE | 4609 | 7.40 | 7.38 | 4996 | 6.58 | 6.58 | 0.82 | 0.000 |

*5.2. Women Directors and Risk Management and Derivative Usage*

Refer to Table 6 in line with the Equations (5) and (6) presented in this section; Model 1 captures the simultaneous equations models for the relationship between derivatives and leverage. In Table 6, Models 2, 3, and 4 are compiled based on Equations (5) and (6) in this section, displaying results for board gender diversity on derivatives and the joint effects of board diversity with investment, liquidity, and financial policies. Model 5 presents results for the entire sample taken together. All models use the simultaneous equations model with debt decisions. The results for models 2, 3 and 4 indicate that board diversity–liquidity, financial, and investment strategies–impacts firms' derivatives decisions and is significant at 1%. While model 5 presents the pooled model taking all decisions to show an increase (decrease) in derivatives use due to financial (investment and liquidity) decisions. The results indicate that women directors actively participate in the derivatives decisions of the firm through their risk monitoring strategies.

$$LEVERAGE_{i,t}^a = b_0 + b_1 DER_{i,t}^b + b_2 RnD_{i,t} + b_3 ROA_{i,t} + b_4 SIZE_{i,t} + b_5 VOL_{i,t} + B_{i,t} \ldots \tag{5}$$

$$\begin{aligned} DER_{i,t}^b = a_0 &+ a_1 BDDIVER_{i,t} + a_2 BDDIVERSLEV_{i,t}^a + a_3 BDDIVERSRD_{i,t} + \\ &a_4 BDDIVERSLIQUIDITY_{i,t} + a_5 LEVERAGE_{i,t}^a + a_6 RnD_{i,t} \\ &+ a_7 LIQUIDITY_{i,t} + a_8 TLC_{i,t} + a_9 MSC_{i,t} \\ &+ a_{10} SIZE_{i,t} + \alpha_{i,t} \ldots \end{aligned} \tag{6}$$

The results presented in Table 7 (Panel A) based on Equation (7) demonstrate that women directors increase firm value, with a larger effect for derivative user firms than non-users. Therefore, the results do not support the null hypothesis and endorse the alternative hypothesis: H2a: Gender heterogeneity on the corporate boards enhances firm value for US-listed firms that use derivatives. The risk models show that gender diversity reduces stock return volatility, see Table 7 (Panel B), in both groups of firms. Still, the effects are larger for derivative users, while board diversity reduces bankruptcy risk, see Table 7 (Panel C), and the effects are only visible for derivative users. Therefore, the results do not support the null hypothesis and endorse the alternative hypothesis: H1a: Gender heterogeneity on the corporate boards reduces risk in US-listed firms that use derivatives.

**Table 6.** Simultaneous equations models for the relationship of leverage, derivatives and corporate governance.

| Variables | (1) Lev coeff | t-stat | (2) Der Coeff | z-stat | (3) Liquidity coeff | z-stat | (4) Invest Coeff | z-stat | (5) Financial coeff | z-stat | Pooled Coeff | z-stat |
|---|---|---|---|---|---|---|---|---|---|---|---|---|
| DERb | 0.35 *** (0.385) | 8.97 | | | | | | | | | | |
| ROA | −0.02 *** (0.002) | −10.08 | | | | | | | | | | |
| SIZE | −0.01 0.007 | −1.15 | | | | | | | | | | |
| VOL | 0.08 *** (0.012) | 6.82 | | | | | | | | | | |
| RnD | −0.12 *** (0.010) | −12.49 | 0.17 *** (0.030) | 5.54 | 0.17 *** (0.030) | 5.62 | 0.08 ** (0.041) | 2.04 | 0.05 *** (0.012) | 4.77 | 0.024 *** (0.014) | 1.78 |
| BDDIVERS | | | 0.02 (0.015) | 1.34 | 0.015 (0.015) | 1.04 | −0.019 0.019 | −10.99 | −0.05 *** (0.005) | −9.40 | −0.08 *** (0.008) | −9.77 |
| BDDIVERS_LEVa | | | | | | | | | 0.21 *** (0.022) | 9.26 | 0.25 *** (0.025) | 10.01 |
| BDDIVERS_RD | | | | | | | 0.09 *** 0.027 | 3.19 | | | 0.03 *** (0.005) | 6.54 |
| BDDIVERS_LIQTY | | | | | 0.07 *** (0.017) | 4.31 | | | | | 0.03 *** (0.004) | 5.90 |
| LEVERAGEa | | | 0.20 (0.176) | 1.12 | 0.18 (0.176) | 0.31 | 0.21 (0.176) | 1.22 | −0.19 ** (0.095) | −1.99 | −0.26 *** (0.098) | −2.61 |
| LIQUIDITY | | | −0.14 *** (0.026) | −6.40 | −0.20 *** (0.026) | −7.66 | −0.14 *** (0.022) | −6.26 | −0.05 *** (0.008) | −6.33 | −0.08 *** (0.011) | −7.28 |
| TLCF | | | −0.092 *** (0.032) | −3.40 | −0.09 *** (0.027) | −3.49 | −0.09 *** (0.027) | −3.42 | −0.01 ** (0.004) | −2.14 | −0.012 *** (0.004) | −2.66 |
| MSC | | | −0.01 (0.045) | −0.13 | 0.01 (0.045) | −0.26 | −0.002 (0.045) | −0.06 | 0.02*** (0.005) | 5.17 | 0.027*** (0.005) | 5.05 |
| SIZE | | | 0.11 *** (0.009) | 13.21 | 0.12 *** (0.009) | 13.38 | 0.11 *** (0.009) | 12.95 | 0.05 *** (0.003) | 15.35 | 0.05 *** (0.003) | 15.27 |
| Constant | 0.06 (0.073) | 0.83 | −0.95 *** (0.071) | −13.32 | −0.94 *** (0.071) | −13.23 | −0.90 *** 0.073 | −12.23 | 0.16 *** (0.030) | 5.45 | 0.21 *** (0.032) | 6.46 |
| Fixed effects | yes | | yes | | yes | | yes | | yes | | yes | |
| Observations | 9605 | | 9605 | | 9605 | | 9605 | | 9605 | | 9605 | |
| Pseudo R2 | 0.35 | | 0.04 | | 0.04 | | 0.04 | | 0.03 | | 0.03 | |

The predicted value from the other equation is denoted by (5) and (6). The *** and ** represent *p*-value significance at 0.01 and 0.05. The definitions variables are available in Table 3.

$$VALUE_{i,t} = h_0 + h_1 BDDIVERS_{i,t} + h_2 LEVERAGE_{i,t} + h_3 RnD_{i,t} + h_4 ROA_{i,t} + h_5 SIZE_{i,t} \\ + h_6 CAPEX_{i,t} + h_7 ZSCORE_{i,t} + \sigma_{i,t} \dots \tag{7}$$

Results indicate that women on boards positively influence performance in both derivative user and non-user firms. Still, the benefits are greater in firms that use derivative instruments. Taken together, it indicates that their risk monitoring impacts result in enhanced value and reduction in both stock return volatility and bankruptcy risk. Our findings support our hypothesis, suggesting that any agency conflicts manifested through the misuse of derivatives are mitigated by gender diversity on board.

**Table 7.** The relationship of value, stock return volatility, bankruptcy risk and corporate governance non-user and user for derivatives.

| Variables | Pooled coeff | t-stat | DER USERS coeff | t-stat | NON USERS Coeff | t-stat |
|---|---|---|---|---|---|---|
| Panel A: Value as a dependent variable | | | | | | |
| BDDIVERS | 0.02 *** (0.005) | 3.64 | 0.022 *** (0.007) | 3.25 | 0.021 ** (0.009) | 2.38 |
| LEVERAGE | −0.75 *** (0.036) | −20.87 | −0.76 *** (0.046) | −16.43 | −0.71 *** (0.054) | −13.22 |
| RnD | 0.15 *** (0.010) | 14.60 | 0.08 *** (0.013) | 5.87 | 0.21 *** (0.015) | 13.78 |
| ROA | 0.03 *** (0.003) | 8.77 | 0.03 *** (0.005) | 6.03 | 0.32 *** (0.004) | 7.11 |
| SIZE | −0.06 *** (0.004) | −17.30 | −0.04 *** (0.005) | −8.03 | −0.07 *** (0.005) | −14.17 |
| CAPEX | 1.00 *** (0.073) | 13.66 | 1.07 *** (0.105) | 10.26 | 0.94 *** (0.106) | 8.88 |
| ZSCORE | 0.23 *** (0.011) | 20.36 | 0.22 *** (0.016) | 13.69 | 0.23 *** (0.015) | 15.11 |

**Table 7.** *Cont.*

| | Pooled | | DER USERS | | NON USERS | |
|---|---|---|---|---|---|---|
| Annual effects | yes | | yes | | Yes | |
| Industrial effects | yes | | yes | | Yes | |
| Constant | 0.39 *** | 11.93 | 0.26 *** | 5.51 | 0.43 *** | 9.46 |
| | (0.033) | | (0.048) | | (0.045) | |
| Observations | 9605 | | 4609 | | 4996 | |
| R2 | 0.43 | | 0.44 | | 0.41 | |
| Panel B: Stock Return Volatility as a dependent variable | | | | | | |
| BDDIVERS | −0.04 *** | −7.79 | −0.04 ** | −5.66 | −0.03 *** | −4.97 |
| | (0.005) | | (0.006) | | (0.006) | |
| LEVERAGE | 0.41 *** | 16.60 | 0.46 *** | 13.27 | 0.35 *** | 9.74 |
| | (0.025) | | (0.035) | | (0.036) | |
| LIQUIDITY | 0.01 *** | 2.72 | 0.03 *** | 3.77 | −0.000 | −0.07 |
| | (0.005) | | 0.008 | | (0.007) | |
| ROA | −0.05 *** | −26.27 | −0.06 *** | −19.25 | −0.05 *** | −18.18 |
| | (0.002) | | (0.003) | | (0.002) | |
| SIZE | −0.09 *** | −29.84 | −0.08 *** | −19.45 | −0.09 *** | −22.09 |
| | (0.003) | | (0.004) | | (0.004) | |
| Year effects | yes | | yes | | yes | |
| Industry effects | yes | | yes | | yes | |
| Constant | 4.35 *** | 229.43 | 4.31 *** | 148.00 | 0.46 *** | 172.54 |
| | (0.019) | | (0.029) | | (0.048) | |
| Observations | 9605 | | 4609 | | 4996 | |
| R2 | 0.39 | | 0.39 | | 0.38 | |
| Panel C: Bankruptcy Risk as a dependent variable | | | | | | |
| BDDIVERS | 0.03 *** | 3.38 | 0.07 *** | 5.36 | −0.002 | −0.16 |
| | (0.009) | | (0.012) | | (0.013) | |
| LEVERAGE | −1.70 *** | −36.45 | −1.65 *** | −24.34 | −1.84 *** | −27.11 |
| | (0.047) | | (0.068) | | (0.068) | |
| LIQUIDITY | −0.03 *** | 17.01 | 0.18 *** | 10.83 | 0.22 *** | 13.19 |
| | (0.007) | | 0.017 | | (0.016) | |
| ROA | 0.13 *** | 26.56 | 0.10 *** | 12.99 | 0.14 *** | 23.50 |
| | (0.005) | | (0.008) | | (0.006) | |
| SIZE | 0.20 *** | −3.78 | −0.04 *** | −3.83 | −0.01 | −1.44 |
| | (0.012) | | (0.011) | | (0.010) | |
| Year effects | yes | | yes | | yes | |
| Industry effects | yes | | yes | | yes | |
| Constant | 1.63 *** | 31.38 | 1.75 *** | 20.56 | 1.55 *** | 23.08 |
| | (0.052) | | (0.029) | | (0.067) | |
| Observations | 9605 | | 4609 | | 4996 | |
| R-squared | 0.44 | | 0.44 | | 0.38 | |

The predicted value from the other equation is denoted by (5) and (6). The *p*-values ***, and **, are significant at 0.01, and 0.05. See Table 3 for definitions of dependent and independent variables.

## 5.3. Women Directors and Investment, Liquidity and Financial Policies to Manage Risk

While the preceding section focuses on gender diversity and its impact on risk management activities, this section examines whether women directors further manage risk through their policies and strategies using data compiled using Equations (8)–(10) as presented in Table 8.

$$
\begin{aligned}
VALUE_{i,t} = a_0 &+ a_1 BDDIVERS_{i,t} + a_2 BDDIVERS - LEV_{i,t} + a_3 BDDIVERS - RD_{i,t} + a_4 \\
BDDIVERS_L LIQUIDITY_{i,t} &+ a_5 LEVERAGE_{i,t} + a_6 RnD_{i,t} + a_7 LIQUIDITY_{i,t} + a_8 CAPEX_{i,t} \\
&+ a_9 ROA_{i,t} + a_{10} SIZE_{i,t} + a_{11} Z - SCORE_{i,t} + \alpha_{i,t} \ldots
\end{aligned}
\tag{8}
$$

$$
\begin{aligned}
STOCKRETURNVOLATILITY_{i,t} = e_0 &+ e_1 BDDIVERS_{i,t} + e_2 BDDIVERS_L EV_{i,t} \\
+ e_3 BDDIVERS - RD_{i,t} + eBDDIVERS - LIQUIDITY_{i,t} &+ e_5 LEVERAGE_{i,t} + e_6 RnD_{i,t} + \\
e_7 LIQUIDITY_{i,t} &+ e_8 ROA_{i,t} + e_9 SIZE_{i,t} + \alpha_{i,t} \ldots
\end{aligned}
\tag{9}
$$

$$
\begin{aligned}
BANKRUPTCYRISK_{i,t} = c_0 &+ c_1 BDDIVERS_{i,t} + c_2 BDDIVERS_L EV_{i,t} \\
+ c_3 BDDIVERS - RD_{i,t} + c_4 BDDIVERS_L LIQUIDITY_{i,t} &+ c_5 LEVERAGE_{i,t} + c_6 RnD_{i,t} + c_7 \\
LIQUIDITY_{i,t} &+ c_8 ROA_{i,t} + c_9 SIZE_{i,t} + \alpha_{i,t} \ldots
\end{aligned}
\tag{10}
$$

**Table 8.** Derivative user and non-user firms' value and corporate governance, along with financial, investment, and liquidity strategies.

| | Pooled | | DER USERS | | NON USERS | |
|---|---|---|---|---|---|---|
| Variables | Coeff | t-stat | coeff | t-stat | Coeff | t-stat |
| **Panel A: Value as dependent variable** | | | | | | |
| BDDIVERS | 0.02 | 1.48 | 0.03 ** | 2.11 | 0.003 | 0.20 |
| | (0.011) | | (0.014) | | (0.015) | |
| BDDIVERS_LEV | −0.04 * | −1.66 | −0.11 *** | −3.27 | 0.023 | 0.58 |
| | (0.027) | | (0.034) | | (0.060) | |
| BDDIVERS_RD | 0.03 *** | 3.11 | 0.04 *** | 3.36 | 0.03 * | 1.84 |
| | (0.010) | | (0.013) | | (0.164) | |
| BDDIVERS LIQUIDITY | −0.01 | −0.91 | −0.01 | −0.51 | 0.001 | 0.08 |
| | (0.007) | | (0.009) | | (0.010) | |
| LEVERAGE | −0.71 *** | −16.40 | −0.66 *** | −11.56 | −0.74 *** | −11.51 |
| | (0.043) | | (0.057) | | (0.064) | |
| RnD | 0.12 *** | 7.57 | 0.03 | 1.44 | 0.19 *** | 8.25 |
| | (0.015) | | (0.020) | | (0.023) | |
| LIQUIDITY | 0.02 | 1.59 | 0.02 | 1.55 | −0.005 | −0.36 |
| | (0.010) | | 0.013 | | (0.015) | |
| CAPEX | 1.00 *** | 13.49 | 1.07 *** | 10.17 | 0.93 *** | 8.75 |
| | (0.074) | | (0.106) | | (0.106) | |
| ROA | 0.03 *** | 8.71 | 0.03 *** | 6.04 | 0.03 *** | 6.95 |
| | (0.003) | | (0.005) | | (0.005) | |
| SIZE | −0.06 *** | −16.13 | −0.04 *** | −7.49 | −0.08 *** | −13.57 |
| | (0.004) | | (0.005) | | (0.006) | |
| ZSCORE | 0.23 *** | 19.37 | 0.22 *** | 13.06 | 0.23 *** | 14.40 |
| | (0.012) | | (0.017) | | (0.016) | |
| Annual effects | Yes | | yes | | Yes | |
| Industrial effects | Yes | | yes | | Yes | |
| Constant | 0.40 *** | 11.35 | 0.25 *** | 5.09 | 0.46 *** | 9.51 |
| | (0.035) | | (0.050) | | (0.048) | |
| Sample | 9605 | | 4609 | | 4996 | |
| R2 | 0.43 | | 0.45 | | 0.41 | |
| **Panel B: Stock Return Volatility as a dependent variable** | | | | | | |
| BDDIVERS | −0.03 *** | −3.18 | −0.04 *** | −3.10 | −0.01 | −1.39 |
| | (0.008) | | (0.012) | | (0.011) | |
| BDDIVERS_LEV | 0.03 | 1.13 | 0.04 | 1.30 | 0.02 | 0.46 |
| | (0.025) | | (0.031) | | (0.038) | |
| BDDIVERS_RD | −0.04 *** | −4.49 | −0.01 | −1.17 | −0.06 *** | −5.33 |
| | (0.009) | | (0.012) | | (0.012) | |
| BDDIVERS_LIQUIDITY | 0.03 *** | 5.18 | 0.02** | 1.99 | 0.04 *** | 5.06 |
| | (0.006) | | (0.009) | | (0.008) | |
| LEVERAGE | 0.37 *** | 11.28 | 0.41 *** | 8.58 | 0.32 *** | 6.92 |
| | (0.033) | | (0.048) | | (0.047) | |
| RnD | 0.01 | 1.11 | −0.01 | −0.29 | 0.035 ** | 2.29 |
| | (0.012) | | (0.018) | | (0.015) | |
| LIQUIDITY | −0.01 | −1.02 | 0.01 | 1.53 | −0.03 ** | −3.07 |
| | (0.007) | | 0.013 | | (0.009) | |
| ROA | −0.05 *** | −26.37 | −0.06 *** | −19.31 | −0.05 *** | −18.21 |
| | (0.002) | | (0.003) | | (0.002) | |
| SIZE | −0.09 *** | −29.28 | −0.08 *** | −19.18 | −0.09 *** | −21.41 |
| | (0.003) | | (0.004) | | (0.004) | |
| Year and Industry effects | yes | | yes | | Yes | |
| Constant | 4.35 *** | 208.76 | 4.32 *** | 135.24 | 4.37 *** | 155.11 |
| | (0.021) | | (0.032) | | (0.028) | |
| Observations | 9605 | | 4609 | | 4996 | |
| R2 | 0.39 | | 0.39 | | 0.39 | |
| **Panel C: Bankruptcy Risk as dependent variable** | | | | | | |
| BDDIVERS | 0.06 *** | 4.47 | 0.11 *** | 5.14 | 0.03 ** | 1.97 |
| | (0.013) | | (0.022) | | (0.016) | |
| BDDIVERS_LEV | −0.17 *** | −4.23 | −0.18 *** | −3.30 | −0.21 *** | −3.49 |
| | (0.040) | | (0.056) | | (0.061) | |
| BDDIVERS_RD | 0.02 | 1.51 | −0.01 | −0.31 | 0.03 | 1.27 |
| | (0.014) | | (0.019) | | (0.021) | |
| BDDIVERS_LIQUIDITY | −0.03 ** | −2.10 | 0.003 | 0.16 | −0.04 ** | −2.40 |
| | (0.012) | | (0.019) | | (0.016) | |
| LEVERAGE | −1.60 *** | −26.13 | −1.51 *** | −16.57 | −1.71 *** | −19.45 |
| | (0.061) | | (0.091) | | (0.088) | |
| RnD | −0.12 *** | −5.82 | −0.05 * | −1.68 | −0.16 *** | −5.58 |
| | (0.021) | | (0.030) | | (0.028) | |
| LIQUIDITY | 0.24 *** | 15.36 | 0.19 *** | 8.15 | 0.27 *** | 13.07 |
| | (0.016) | | (0.023) | | (0.021) | |
| ROA | 0.12 *** | 25.94 | 0.10 *** | 12.83 | 0.14 *** | 22.78 |
| | (0.005) | | (0.008) | | (0.006) | |
| SIZE | −0.03 *** | −3.80 | −0.04 *** | −3.82 | −0.02 | −1.62 |
| | (0.007) | | (0.011) | | (0.010) | |

**Table 8.** *Cont.*

|  | Pooled |  | DER USERS |  | NON USERS |  |
|---|---|---|---|---|---|---|
| Variables | Coeff<br>Pooled | t-stat | coeff<br>DER USERS | t-stat | Coeff<br>NON USERS | t-stat |
| Year and Industry effects | yes |  | yes |  | yes |  |
| Constant | 1.66 ***<br>(0.054) | 30.69 | 1.73 ***<br>(0.087) | 19.95 | 1.60 ***<br>(0.072) | 22.40 |
| Observations | 9605 |  | 4609 |  | 4996 |  |
| R-squared | 0.45 |  | 0.44 |  | 0.39 |  |

The *p*-values ***, ** and * are significant at 0.01, 0.05 and 0.001.

Table 8: According to Panel A, women directors have a positive impact on a company's performance by making smart decisions regarding debt and investments. This includes reducing leverage and increasing investments, which ultimately enhances the value of the company. These effects are only noticeable in derivative-user firms. Therefore, the research does not support the null hypothesis and instead endorses the idea that gender diversity on corporate boards improves value through investment and financial-related strategies for US-listed companies. However, the study found no significant impact on liquidity-related strategies, suggesting that gender diversity does not have a significant effect on a company's value when it comes to cash management strategies. The findings from Panel B indicate that the investment and liquidity decisions made by female directors have an impact on the volatility of stock returns. These decisions help to decrease the risk of stock volatility for both derivative users and non-users. However, while female directors' liquidity decisions lead to a reduction in stock volatility for derivative users, it shows an increased risk in the non-user group. In summary, the research results reject the null hypothesis and support the alternative hypothesis that the presence of gender diversity on corporate boards can mitigate risks through investment, financial, and liquidity-related strategies for companies listed in the US. Panel C presents results for bankruptcy risk where the increase in Z-Score indicates a risk reduction. The leverage-reducing strategy increases Z-Score to decrease bankruptcy risk in all the pooled, derivative-user and non-user groups. The investment decisions increase Z-Score (reduce bankruptcy risk) in all three groups. While the liquidity strategy to manage cash reserves results in reduced bankruptcy risk overall and in the derivative user groups.

## 6. Discussion

Our research, which draws upon the UET and Lehman Sisters Hypothesis, has found that female directors are skilled risk strategists for US-listed companies during times of crisis. The listed firms experience more significant value effects when they have female directors with higher representation on the corporate board. Such a difference in the firm's performance is because those female directors have differences in knowledge, experience, and values as compared with men board members, which is the support of UET for our study (e.g., [15,19]. Moreover, firms avoid bias by including a woman's voice in financial decision-making confirming the Lehman Sister Hypothesis assumption. Under the framework of UET, our results indicate higher participation of female directors on boards, ultimately leading to a better firm performance by avoiding bankruptcy and using hedging. Our results are consistent with prior studies (e.g., [11,12,38]) and support the economic benefits brought to the firm by gender-diverse boards (e.g., [9,34]).

## 7. Conclusions

### 7.1. Key Findings of the Research

Board Diversity and Leverage: Model 4 proposes utilizing derivatives to decrease dependence on expensive external borrowing as part of a board diversity strategy. This approach aims to enhance financial capacity and streamline cash flow management. The increased usage of derivatives is a key aspect of this strategy.

Board Diversity and RnD: Model 3 explains how board diversity can enhance investment strategies. It has been observed that increasing the use of derivatives helps ensure smooth cash flow, capturing positive net present value investments and protecting the firm's investments. Having more women on the corporate board also contributes to this positive effect.

Board Diversity and Liquidity: Board Diversity liquidity strategy results show the firm's intention to increase derivative usage, which could smoothen cash flows and enable higher liquidity for operations and investments.

Board Diversity and Leverage: Financial Leverage Strategy: It has been observed that companies with more women on corporate boards tend to increase their use of derivatives, such as futures contracts, options contracts, and credit default swaps, as part of their financial strategy. This increase in hedging is aimed at balancing the reduced borrowing to save on external costs and reaping the benefits of hedging. Therefore, this study interprets the negative signs for leverage (Table 8) as a reduction in the usage of leverage, providing an increase in value. This aligns with the derivative usage to decrease reliance on external costlier debt. Hence, scholars in this study interpreted it as a reduction of financial leverage increases in firm value; in Panel B (Stock Return Volatility), reduction of leverage (debt) provides an increase in stock return; and in Panel C (Bankruptcy risk), the reduction of leverage results in an increase in the Z-Score, i.e., reduction of bankruptcy risk. The increase in stock return volatility may reflect the reduced debt (borrowings) and reduced benefit of debt income tax shield benefits.

Board Diversity and RnD: Investment Strategy: The Board Diversity investment strategy (Model 3) is to increase derivative usage to ensure smooth cash flow to capture positive NPV investments and/or protect their investments. This strategy increases firm value, reduces stock return volatility risk, and increases the Z-Score, reducing the bankruptcy risk.

Board Diversity and Liquidity: Liquidity Strategy: Liquidity strategy results show the board's intention to increase derivative usage, enabling higher liquidity for operations and investments by listening to women members. This strategy increases firm value, reduces stock return volatility risk, and increases the Z-Score, reducing bankruptcy risk to ensure sustainable performance for the company Tables 9 and 10.

**Table 9.** Summary of results—derivative usage as dependent variable.

| Variable | Variable Taken Separately | | | | Pooled |
|---|---|---|---|---|---|
| | Panel A | Panel B | Panel C | Panel D | Panel E |
| Board Diversity | NS | | | | |
| Board Diversity Leverage | | | | + | + |
| Board Diversity RnD | | | + | | − |
| Board Diversity Liquidity | | + | | | − |

NS denotes Not Significant. '+' means positive relationship and '−' means negative relationship;.

Interpreting Coefficients—an example

**Example 1.** *Table 6, Model 5, Pooled (all variables)*

$$BDDIVERS + BDDIVERS - LEV = -0.08 * * * + 0.25 * ** = +0.17(positive)$$

**Table 10.** Summary of results—value and risk models.

| | Value (Panel A) | | | Stock Return Volatility (Panel B) | | | Bankruptcy Risk (Z-Score) (Panel C) | | |
|---|---|---|---|---|---|---|---|---|---|
| | Pooled | Der User | Non User | Pooled | Der User | Non User | Pooled | Der User | Non User |
| Board Diversity | + | + | + | − | − | − | + | + | NS |
| Board Diversity Leverage | − | − | NS | − | − | NS | − | − | − |
| BoardDiversity RnD | + | + | + | − | − | − | + | + | + |
| Board Diversity Liquidity | NS | + | NS | Nulled Effect | − | + | + | + | − |

NS denotes Not Significant; '+' means positive relationship and '−' means negative relationship (Ref. Results Tables 7 and 8).

Interpreting Coefficients—an example (a) Taken from Table 7 Board Diversity Single Effect. (b) Taken from Table 8 Joint Effects. The coefficient for joint effect needs to be read along with the coefficient for board diversity. Example (Table 8, Panel C Bankruptcy Risk, Column 2, Pooled):

$$BDDIVERS - LEV + BDDIVERS = -0.17 * ** + 0.11 * ** = net\ effect\ is -0.06(negative).$$

$$BDDIVERS - LIQUIDITY + BDDIVERS = -0.03 * * + 0.06 * ** = +0.03(positive)$$

In general, the evidence points towards women directors being skilled in utilizing financial, investment, and liquidity strategies to great effect. Researchers have observed that gender-diverse boards tend to use derivatives more often as part of their financial and investment strategies.

Testing a Monotonic Association of Gender Diversity with Firm Risk and Value Effects: There is a possibility that the corporate governance responses may vary depending on firm characteristics, and therefore, a non-monotonic association of governance with firm performance may exist. For instance, corporate governance may only respond to very high levels of risk within firms or exhibit restraint when faced with high levels of risk. This study conducts regression analyses to examine whether there is any non-monotonicity in women directors' behavior to risk and value [48]. The technique enables us to investigate board diversity responses at different quantiles of firm performance. The quantile regression analyses with value, stock return volatility, and bankruptcy risk are depicted in Figures 1 and 2. The figures show that the direction of the relationships does not change, demonstrating a monotonic effect of women directors' financial, investment and liquidity policies with the board's diversity to ensure the company's sustainable performance.

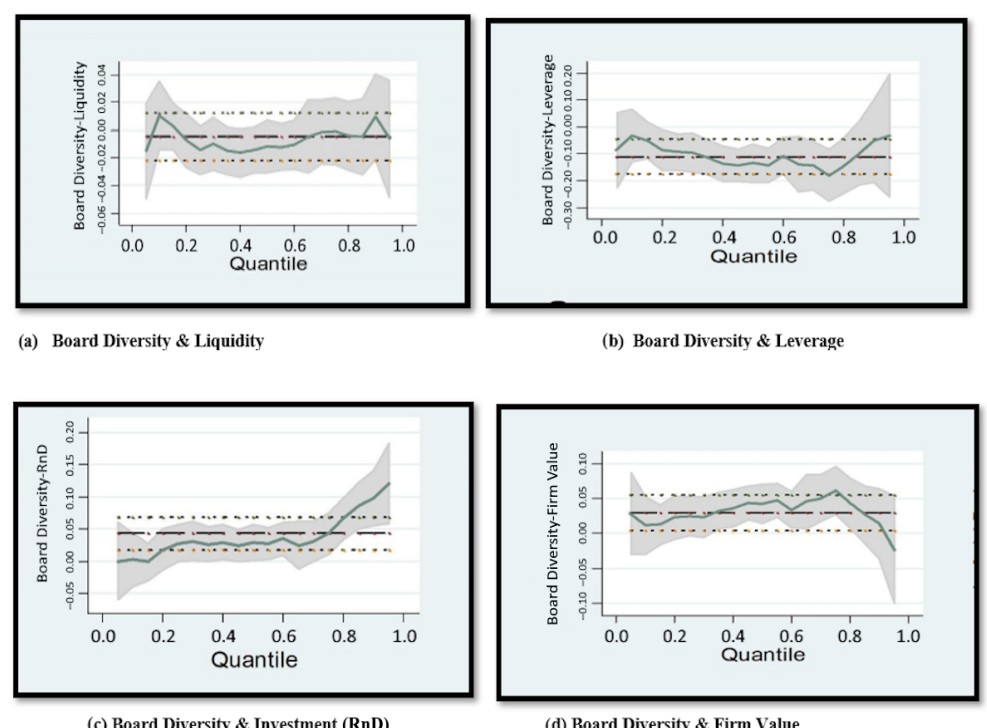

**Figure 1.** Quantile regression analyses of women directors and firm value.

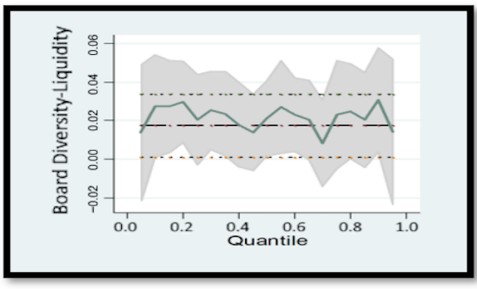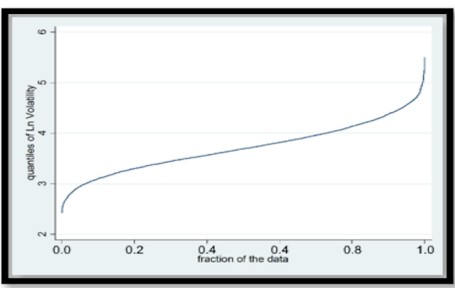

**(a) Women Directors and Stock Return Volatility**

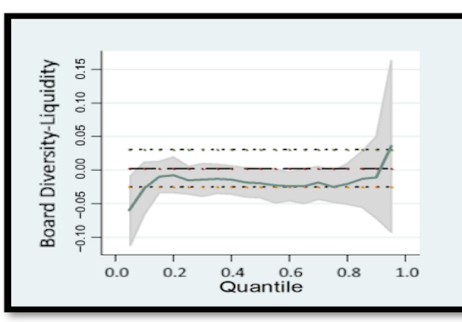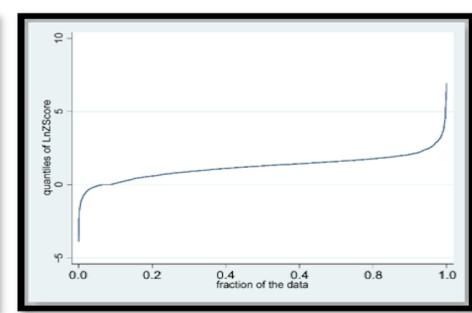

**(b) Women Directors and Bankruptcy Risk**

**Figure 2.** Quantile regression analysis of women directors, stock return volatility and bankruptcy risk.

### 7.2. Practical Implications

Our study has practical implications for corporate boards. First, a unique innovation in our model is that when derivative-user firms are viewed in isolation, results show that women monitor the hedging activities of the firm through their impacts on the determinants of derivatives related to the firm's investment, financial and liquidity policies. So, this study supported a UET-based theoretical model where knowledge, skills and abilities come from board diversity, creating unique cognitive frames that may positively enhance the corporate board's decision-making process. So, female directors actively participate in the firm's financial strategies by increasing derivatives use and reducing reliance on the more expensive external debt financing, resulting in a higher firm's value and reduced bankruptcy risk. These findings are consistent with preceding research in a similar area [28,42]. Second, towards the management of the liquidity needs of the firm, women on board increase cash reserves and control derivatives usage to achieve reduced stock return volatility risk (e.g., [32]. At the same time, board diversity deters derivatives usage and increases value-enhancing investments as part of their investment strategies. The third overarching finding of the present study is that corporate boards hiring females seem to benefit from the lowered risk of bankruptcy. These women directors can be leveraged for better financial strategies, which is important for developing a corporate board's culture where minority voices are considered and heard. For corporations operating in industries with limited potential female directors, efforts should be made to attract female directors in ways that may capture some of their unique characteristics and advantages. In addition, the ability of the firm to attract and retain female directors is likely to depend on the socio-cultural context and the confidence in their judgment and decisions on the board.

### 7.3. Limitations and Future Research Directions

This study is not without limitations and is based on those limitations. Therefore, the study recommends future research to examine Lehman Sister's Hypothesis in line with social-emotional wealth for family firms, banks and entrepreneurs [4,6,8,33,34]. The present study uses a secondary dataset. So, in the future, scholars can rely on primary data in a given context to bring unique evidence, including family firms using financial

derivatives [9,49,50]. Like many other studies, this study relied on data from NYSE to test our model, and the first-hand experience of women directors for the corporate board's decision-making process was missing. Thus, in the future, the study recommends a mixed method using qualitative data and meta-analyses to address this limitation and capture how female directors' cognitive frames shape the firm's strategic decisions, seek and interpret information and financial decision-making processes [4,34,41,50]. The current study did not investigate indirect or moderating effects. However, future research could delve into potential moderating factors, such as gender equality and shareholder protection levels. These factors may impact the connection between female directors and strategic decision-making [20,24]. Upper Echelons Theory suggests that board members, including minority groups like females, are likelier to bring unique values and contribute through distinct experience and knowledge. So, in the future, focus group interviews of the monitoring group directors can help to understand whether boards require a critical mass of two or more female board members before they can effectively leverage strategic decision-making through values, knowledge, and experience to ensure sustainable performance [19,20,51]. Our results claim that firms' female directors positively impact strategic choices for better financial returns. So, scholars can continue to examine whether female directors have a similar positive impact on the strategic decision when a director's appointment is not a representation of tokenism in the future using socioemotional wealth theory and family business context ([33,50]. Lastly, the board's gender diversity–performance nexus lies in the difficulty of accounting for endogeneity caused by omitted variables bias and reverse causality [24,34]. So, to expand the scope of scholarships in corporate governance, scholars can address the endogeneity issue in their future research [5,20,51].

This study concludes that women are active risk managers, formulate effective risk control strategies and are more effective within firms that use derivatives instruments. They manage derivatives usage, control debt utilization, enhance investments and build cash reserves to achieve hedging value premium and risk reduction. The study endorses that women are effective risk managers but have larger impacts in a derivatives environment. A fractionally larger number of women are on boards of derivative user firms. Scholars of this study could attribute the difference in results for the two groups to the lesser influence of women in non-user firms and the smaller corporate board size. According to the findings of this research, having women directors on the board is crucial for effective risk management. It is recommended that they are allowed to participate in financial, investment, and liquidity strategies if the company frequently deals with common derivatives such as futures, forwards, swaps, and options.

**Author Contributions:** Conceptualization, R.T.; methodology and data analysis R.T.; data collection: U.N.; revision and proofreading: I.S.; correspondence and funding: M.A.; revision and proofreading: M.S. All authors have read and agreed to the published version of the manuscript.

**Funding:** We are grateful to the management of the International University of Applied Sciences in Germany for covering the cost of APC or open-access publication for this paper.

**Data Availability Statement:** Not applicable.

**Acknowledgments:** We acknowledge the support of the supervisor (Tony van Zijl) during the PhD of Robina Tashfeen from Victoria University of Wellington Kelburn, Wellington 6012, New Zealand for data collection during PhD from sources including Bloomberg, Compustat and other databases).

**Conflicts of Interest:** The authors declare no conflict of interest.

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
