# Peer review of "How Do Women on Board Reduce a Firm’s Risks to Ensure Sustainable Performance during a Crisis?"

_sustainability, doi:10.3390/su151411145_

Round 1
Reviewer 1 Report
Abstract is not bright and does not attract the reader's attention. Please correct it.
Introduction needs to be improved. It should include more things that highlight why the issue addressed in the article is important, you should give there the structure and purpose of the article. All this is missing.
Add highlights
The literature review itself is unclear. You do not show the search criteria and the state of knowledge about the topic you are talking about. You start on line 58 and end on line 147. You are only showing examples of articles on this topic, while there are many more. You're adding hypotheses that I don't quite understand. You have to correct that
In Chapter 5, you present the formula and then the tables. On what basis are the tables calculated this statistical test? It's not clear. This needs to be described in more detail. So are the rest of the tables. You don't refer to them in the text at all. What are they for?
The methodology is not clear. Include specific steps there. Now it's very broken and I don't know what to do with it.
The results are not well described. Correct them by referring to the tables and what they contain.
Discussion and Conclusions should be written as separate chapters.
Discussion should be expanded and corrected now it doesn't say much about what is in the article.
Conclusions must be improved! They're too short. Consider whether you should move some paragraphs from the discussion to conclusions.
Present more broadly the directions of further research.
Author Response
Dear Reviewer,
Thank you for giving us valuable suggestions to improve the article.
We have incorporated the changes.
The abstract is not bright and does not attract the reader's attention. Please correct it.
Reply: updated
The introduction needs to be improved. It should include more things that highlight why the issue addressed in the article is important. You should give there the structure and purpose of the article. All this is missing.
Reply: Two paragraphs have been added to address the concern of the Reviewer.
Add highlights
The literature review itself is unclear. You do not show the search criteria and the state of knowledge about the topic you are talking about. You start on line 58 and end on line 147. You are only showing examples of articles on this topic, while there are many more. You're adding hypotheses that I don't quite understand. You have to correct that.
Reply: The theoretical parts and two new paragraphs have been added to address this concern and now the hypothesis is more justified.
In Chapter 5, you present the formula and then the tables. On what basis are the tables calculated for this statistical test? It's not clear. This needs to be described in more detail. So are the rest of the tables. You don't refer to them in the text at all. What are they for?
Reply: The more details are presented. See related tables
The methodology is not clear. Include specific steps there. Now it's very broken, and I don't know what to do with it.
Reply: After proofreading, The method part is now more clear.
The results are not well described. Correct them by referring to the tables and what they contain.
Reply: The tables are now correctly referred to. But due to the formatting of the overleaf format of MDPI. You may find those tables being presented a little misplaced. So this issue will be addressed once the journal subeditors finalize the paper.
Discussion and Conclusions should be written as separate chapters.
Reply: separated, and subjections are added.
Discussion should be expanded and corrected now. It doesn't say much about what is in the article.
Conclusions must be improved! They're too short. Consider whether you should move some paragraphs from the discussion to conclusions.
Reply: modified
Present more broadly the directions of further research.
Reply added:
Thanks
Reviewer 2 Report
1. Introduction- Please provide more context. Begin by briefly introducing the current state of the debate on women's effectiveness in managing firm performance and risk. This will set the stage for the subsequent discussions and arguments.
2. Introduction- Please strengthen the literature review. While you have mentioned a range of contrasting viewpoints on the link between women on boards and firm performance, it would be beneficial to provide a more comprehensive and balanced overview of the existing literature. Highlight key studies, their findings, and the reasons behind conflicting results. This will establish a solid foundation for the research gap you intend to address.
3. Introduction- Please clarify the Lehman Sisters Hypothesis. Explain the hypothesis in more detail, including the rationale behind the assertion that the global financial crisis could have been averted if women had led Lehman Brothers. This will enhance the readers' understanding of the hypothesis and its significance.
4. Discussion and conclusions should not be placed in one section.
Moderate editing of English language required
Author Response
Dear Reviewer,
Thank you for giving us valuable suggestions to improve the article.
We have incorporated the changes.
- Introduction- Please provide more context. Begin by briefly introducing the current state of the debate on women's effectiveness in managing firm performance and risk. This will set the stage for the subsequent discussions and arguments.
Reply: the intro has now followed the classical approach of four paragraphs to define the problem, followed by brief literature, gaps/significance and the research objectives.
- Introduction- Please strengthen the literature review. While you have mentioned a range of contrasting viewpoints on the link between women on boards and firm performance, it would be beneficial to provide a more comprehensive and balanced overview of the existing literature. Highlight key studies, their findings, and the reasons behind conflicting results. This will establish a solid foundation for the research gap you intend to address.
Reply: the details are provided in the development of the hypotheses section.
- Introduction- Please clarify the Lehman Sisters Hypothesis. Explain the hypothesis in more detail, including the rationale behind the assertion that the global financial crisis could have been averted if women had led Lehman Brothers. This will enhance the readers' understanding of the hypothesis and its significance.
Reply: Details about the Lehman Sisters Hypothesis are available in the literature review section 2.1.
- Discussion and conclusions should not be placed in one section.
Reply:
Sections have been separated, and subheadings have been properly added to make them more readable. Now two sections are also in line with the comments of reviewer 3.
Thanks
Reviewer 3 Report
· The pronoun “We” may be replaced by the authors/researchers/study.
· In lines number 3 and 4, “The analysis is based on a unique dataset of US-listed firms - 3 derivative User and Non-User listed - to” Is it hyphen or dash? Please use it accordingly. There are a lot of misusages of hyphens and dash. Kindly correct the errors throughout the manuscript.
· Introduction is well-defined.
· The hypotheses development using literature is flawless. However, the authors may provide the literature gap to execute the research.
· Please mention the data collection years.
· Methodology and Results are clearly portrayed.
· Tables 3 and 6 can be customized for better reading visibility.
· A separate section for Discussion can be given as it is presently clubbed with the conclusion. Besides, the authors may compare and contrast the present study findings with existing similar research findings.
· Further, the Conclusion section may be framed as follows,
Practical Implications
Key findings of the research
Limitations and Future Research Directions
· References more than ten years old may be replaced with new ones.
· The language of the paper may be polished.
· The following article can be cited in the revised manuscript as it may relate more to the present study.
§ Ingalagi, S. S., Nawaz, N., Rahiman, H. U., Hariharasudan, A., & Hundekar, V. (2021). Unveiling the crucial factors of women entrepreneurship in the 21st century. Social Sciences, 10(5) doi:10.3390/socsci10050153
§ Georgieva, S. (2022). Women's Entrepreneurship – Segmentation And Management Dimensions. Polish Journal of Management Studies, 26(2), 144-161. doi:10.17512/pjms.2022.26.2.09
§ Åšlusarczyk, B., & Broniszewska, A. (2014). Entrepreneurship of women in Poland and the EU - quantitative analysis. Polish Journal of Management Studies, 9, 217-224.
The language of the paper may be polished.
Author Response
Dear Reviewer,
Thank you for giving us valuable suggestions to improve the article.
We have incorporated the changes.
- The pronoun “We” may be replaced by the authors/researchers/study.
Reply:
- In lines number 3 and 4, “The analysis is based on a unique dataset of US-listed firms - 3 derivative User and Non-User listed - to” Is it hyphen or dash? Please use it accordingly. There are a lot of misusages of hyphens and dash. Kindly correct the errors throughout the manuscript.
Reply: done
- Introduction is well-defined.
Reply: Thanks
- The hypotheses development using literature is flawless. However, the authors may provide the literature gap to execute the research.
Reply: see paragraph 2 of the introduction section.
- Please mention the data collection years.
Reply: mentioned in the method section.
- Methodology and Results are clearly portrayed.
Reply: Thanks
- Tables 3 and 6 can be customized for better reading visibility.
Reply: We have used Overleaf and tried our best to format the tables/images on paper. For instance, table 8 is automatically moving to the discussion section, which is beyond our control. I hope the journal will address the visibility issues by editing latex code and adjusting long tables in the particular space.
- A separate section for Discussioncan be given as it is presently clubbed with the conclusion. Besides, the authors may compare and contrast the present study findings with existing similar research findings.
Reply: added
- Further, the conclusion section be framed as follows,
Practical Implications
Key findings of the research
Limitations and Future Research Directions
Reply: The headings and related body have been added, and the conclusion section has been framed accordingly.
- References more than ten years old may be replaced with new ones.
Reply: The list has been updated by removing more than 15 old references. However, key theory references can not be removed.
- The language of the paper may be polished.
Reply: We have proofread the paper through a premium version of Grammarly.
- The following article can be cited in the revised manuscript as it may relate more to the present study.
- Ingalagi, S. S., Nawaz, N., Rahiman, H. U., Hariharasudan, A., & Hundekar, V. (2021). Unveiling the crucial factors of women entrepreneurship in the 21st century.Social Sciences, 10(5) doi:10.3390/socsci10050153
- Georgieva, S. (2022). Women's Entrepreneurship – Segmentation And Management Dimensions. Polish Journal of Management Studies, 26(2), 144-161. doi:10.17512/pjms.2022.26.2.09
- Åšlusarczyk, B., & Broniszewska, A. (2014). Entrepreneurship of women in Poland and the EU - quantitative analysis. Polish Journal of Management Studies, 9, 217-224.
Reply:
The latest references have been added.
Round 2
Reviewer 1 Report
The authors took into account the comments I asked for.
Author Response
The authors took into account the comments I asked for.
Reply: Thanks
Reviewer 2 Report
Thank you for your revision.
Moderate editing of English language required
Author Response
Comment: Moderate editing of English language required
Redply. Proof reading done.
Reviewer 3 Report
· For this earlier comment, "The pronoun ‘We’ may be replaced by the authors/researchers/study". The authors have not corrected it appropriately. See the lines – 470, 390, etc.
· The provided literature gap is not sufficiently supported to execute the research. By the way, the literature gap must be given after reviewing the literature. Here, it is mentioned at the bottom of the introduction section and before the literature. Revise it accordingly. The gap must be identified from the literature, not by the authors’ perception.
· Please mention the data collection years – for this comment; the authors have not carried out anything. In the manuscript, the years of data collection are not given. In addition, the authors’ responses are not clear. They have just given a short note on every comment. Please mention the line numbers, paragraphs or clues to identify the corrections carried out.
· The present form of Tables 3 and 6 cannot be read. It should be revised.
· The discussion section is poorly constructed. It should be improved, as mentioned earlier.
· The language of the paper must be improved. 'Grammarly' is an AI support tool. We should completely depend on it. See, for example, line no 33 – “to address latest cal for the research”. Is it “call for” or “cal for”? and the latest or latest? The paper must be language corrected or checked by a native speaker of English or a Professional English Language Editor.
The language of the paper must be improved. 'Grammarly' is an AI support tool. We should completely depend on it. See, for example, line no 33 – “to address latest cal for the research”. Is it “call for” or “cal for”? and the latest or latest? The paper must be language corrected or checked by a native speaker of English or a Professional English Language Editor.
Author Response
Comment: For this earlier comment, "The pronoun 'We' may be replaced by the authors/researchers/study". The authors have not corrected it appropriately. See the lines – 470, 390, etc.
Reply: The issue has been addressed with professional proofreading.
Comment: The provided literature gap is not sufficiently supported to execute the research.
Reply: We do not agree with these comments. Research gaps have been properly referenced and have strong theoretical and empirical papers support. Please take some time to read the referred articles.
Comment: By the way, the literature gap must be given after reviewing the literature. Here, it is mentioned at the bottom of the introduction section and before the literature. Revise it accordingly. The gap must be identified from the literature, not by the authors' perception.
Reply: We do not agree to move this paragraph of the paper from an intro to a literature section. This is normal practice for papers to research gaps as a part of the intro. The practice reviewer is suggesting is for the thesis, not for the research articles. Otherwise, the introduction will be unable to attract the reader attraction
Comment: Please mention the data collection years – for this comment, the authors have not carried out anything. In the manuscript, the years of data collection are not given.
Reply: year added (see data and sample).
In addition, the authors' responses are not clear. They have just given a short note on every comment. Please mention the line numbers, paragraphs or clues to identify the corrections carried out.
Reply: the modified section was correctly highlighted.
Comment: The present form of Tables 3 and 6 cannot be read. It should be revised.
Reply: the final formatting shall be done by the journal. We have tried our best as we are using overleaf/latex to format, so there are some limitations.
Comment: The discussion section is poorly constructed. It should be improved, as mentioned earlier.
Reply: the discussion was revised as guided. Proper headings were added, and subsections were added and highlighted.
Comments: The language of the paper must be improved. 'Grammarly' is an AI support tool. We should completely depend on it. See, for example, line no 33 – "to address latest cal for the research". Is it "call for" or "cal for"? and the latest or latest? The paper must be language corrected or checked by a native speaker of English or a Professional English Language Editor.
Reply: a native speaker of English or a Professional English Language Editor was hired for proofreading from our English langue department.
Thanks for your comment.
Round 3
Reviewer 3 Report
The authors have ignored/do not agree with some of my comments. So, I reject this paper.
The language of the paper must be improved. 'Grammarly' is an AI support tool. We should not completely depend on it. See, for example, line no 33 – "to address latest cal for the research". Is it "call for" or "cal for"? and the latest or latest? The paper must be language corrected or checked by a native speaker of English or a Professional English Language Editor. The certificate of proofreading and editing can be submitted.
The authors mentioned that they have hired a language editor for proofreading their paper but still, the mentioned error persists.